# Presepsin Does Not Predict Risk of Death in Sepsis Patients Admitted to the Intensive Care Unit: A Prospective Single-Center Study

**DOI:** 10.3390/biomedicines12102313

**Published:** 2024-10-11

**Authors:** Michał P. Pluta, Piotr F. Czempik, Magdalena Kwiatkowska, Katarzyna Marczyk-Bełbot, Sebastian Maślanka, Jolanta Mika, Łukasz J. Krzych

**Affiliations:** 1Department of Acute Medicine, Medical University of Silesia, 41800 Zabrze, Poland; anestitsos@sum.edu.pl; 2Department of Cardiac Anesthesia and Intensive Therapy, Silesian Center for Heart Diseases, 41800 Zabrze, Poland; 3Department of Anesthesiology and Intensive Therapy, Medical University of Silesia, 40752 Katowice, Poland; pczempik@sum.edu.pl; 4Students’ Scientific Society “#Intensywna_Po_Godzinach”, Department of Acute Medicine, Medical University of Silesia, 41800 Zabrze, Poland

**Keywords:** sepsis, presepsin, biomarker, prediction

## Abstract

**Background**: Sepsis is defined as life-threatening organ dysfunction caused by an abnormal host response to infection. The study aimed to evaluate the utility of presepsin (P-SEP) in predicting the risk of death in patients with sepsis at the time of intensive care unit (ICU) admission. **Methods**: Adult patients were included in the study if they met SEPSIS-3 criteria at ICU admission. Demographic and clinical data were collected. The following inflammatory parameters were determined: C-reactive protein (CRP), procalcitonin (PCT), interleukin-6 (IL-6), and presepsin (P-SEP). Material was collected for microbiological testing depending on the suspected source of infection. The primary endpoint was patient death before ICU discharge. The secondary endpoint was a positive microbiological test result. **Results**: Eighty-six patients were included in the study. Thirty patients (35%) died before discharge from the ICU. There was no difference in P-SEP, CRP, PCT, and IL-6 values between patients who survived and those who died (*p* > 0.05 for all). P-SEP, CRP, PCT, and IL-6 were determined at ICU admission and did not accurately predict the risk of death in ROC curve analysis (*p* > 0.05 for all). Confirmation of the location of the focus of bacterial infection by microbiological testing was obtained in 43 (49%) patients. P-SEP, PCT, CRP, and IL-6 were significantly higher in patients with positive microbiological findings. **Conclusions**: In patients with suspected sepsis admitted to the Intensive Care Unit, presepsin does not accurately predict the risk of in-hospital death, but it can predict a positive microbiological culture.

## 1. Introduction

Sepsis is defined as life-threatening organ dysfunction caused by an abnormal host response to infection [1]. However, differentiating sepsis from non-infectious organ failure (NIOF) early in treatment is difficult [2]. Unwarranted use of broad-spectrum antibiotic therapy in NIOF promotes antibiotic resistance [3], but every hour of delayed antibiotic therapy in bacterial sepsis significantly worsens prognosis [1]. Many areas of medicine are looking for specific and sensitive biomarkers for early-stage disease screening to facilitate the diagnostic and therapeutic process. In the case of sepsis, the ideal biomarker should predict the infectious (primarily bacterial) etiology of organ failure and correlate with the severity of the disease, identifying patients at higher risk of death and in whom antimicrobial treatment will be effective. This would allow better use of limited intensive care unit (ICU) resources and rational antibiotic management [4].

Of the markers used clinically, C-reactive protein (CRP) shows high diagnostic sensitivity but has low specificity for bacteremia. Interleukin-6 (IL-6) has moderate accuracy in differentiating sepsis from NIOF [5]. Procalcitonin (PCT), on the other hand, can increase in many non-septic conditions [6]. However, in confirmed sepsis, the lack of a decrease in PCT in the first four days of treatment is an unfavorable prognostic indicator [7]. Presepsin (P-SEP) is a newer biomarker that is potentially more specific for bacterial infection. The transmembrane glycoprotein mCD14 localized on the surface of monocytes and macrophages and the soluble glycoprotein sCD-14 bind bacterial lipopolysaccharides (LPS) on the surface of gram-negative bacteria in the presence of lipoprotein-binding protein (LBP). The resulting CD14-LPS-LBP complex, under the influence of cathepsin D, causes the release of the N-terminal fragment of the soluble form of CD14 (sCD14-ST), known as P-SEP, into the blood [8]. In healthy young volunteers without inflammatory disease, P-SEP values do not exceed 200 pg/mL [6] and increase up to threefold in healthy patients over 70 years of age [9] and with kidney damage [6]. Several studies have reported a cutoff point >0.6 ng/mL for predicting bacterial infection [10].

The purpose of this study was to evaluate the utility of P-SEP in predicting the risk of death in patients with sepsis at the time of ICU admission. It was also planned to compare whether P-SEP predicts death better than standardly determined parameters: WBC, IL-6, CRP, and PCT. Additionally, it was planned to compare the values of inflammatory markers in a group of patients with positive and negative microbiological tests for bacterial infections.

## 2. Materials and Methods

### 2.1. Study Design

This prospective study was conducted in the medical intensive care unit (ICU) of a multi-profile university hospital between 01/2021 and 12/2022. Positive approval was obtained from the bioethics committee for the study (PCN/CBN/0022/KB1/27/I/21). Determinations of inflammatory markers were made from blood samples collected for routine laboratory tests performed upon admission to the ICU, which did not expose the patient to additional blood loss and did not require additional informed consent. The P-SEP determination result was unknown to the attending physician and therefore did not influence clinical decisions.

### 2.2. Patients

Adult patients were included in the study if they met SEPSIS-3 criteria at ICU admission in the physician’s judgment [11]. According to the Surviving Sepsis Campaign guidelines, sepsis was defined as life-threatening organ dysfunction caused by a dysregulated host response to infection. Organ dysfunction was identified by an acute change in the Sequential Organ Failure Assessment (SOFA) total score by ≥2 points following a suspected infection.

### 2.3. Clinical and Laboratory Data

Basic clinical and demographic data were collected from electronic patient records (AMMS, Asseco Medical Solution, Rzeszow, Poland). The degree of organ failure and predicted risk of death were determined using the APACHE II (Acute Physiology And Chronic Health Evaluation), SAPS II (Simplified Acute Physiology Score), and SOFA (Sequential Organ Failure Assessment) qualification systems.

Venous blood samples were collected in dedicated tubes (BD Vacutainer, Becton Dickinson, Swindon, UK) and transferred immediately to the hospital laboratory. Hematological parameters were determined using an XN-1000 analyzer (Sysmex, Kobe, Japan) and CRP levels were measured with a CobasPRO instrument (Roche Diagnostics GmbH, Vienna, Austria). PCT and IL-6 were determined by electrochemiluminescence (ECLIA) using Elecys reagents (Roche Diagnostics GmbH, Vienna, Austria). R&D System/Biotechne (Minneapolis, MN, USA) reagents were used to determine P-SEP. In each patient, blood was drawn for microbiological testing on aerobic and anaerobic media. In addition, depending on the suspected location of the infection, urine, cerebrospinal fluid, body cavity fluid, bronchoalveolar lavage, or tracheal aspirate, samples were collected to confirm the location of the focus of the infection definitively.

### 2.4. Statistical Analysis

Statistical analysis was performed using MedCalc 18 software (MedCalc Software bvba, Ostend, Belgium). Quantitative variables were presented as medians and interquartile ranges (IQR). Qualitative variables were presented as absolute values and percentages. The distribution of variables was verified using the D’Agostino-Pearson test. Differences between quantitative variables were assessed using ANOVA or the Kruskal–Wallis test, depending on the distribution of the variables. Diagnostic accuracy was assessed using ROC curves and area under the curve (AUC). A value of *p* < 0.05 was considered statistically significant.

### 2.5. Outcomes

The primary endpoint was patient death before ICU discharge. The secondary endpoint was a positive microbiological test result.

## 3. Results

In total, 86 patients were included in the study, including 48 men (56%). The median age was 65 years (IQR 51–72). Thirty patients (35%) died before ICU discharge. Patients who died had higher APACHE II (25, IQR 19–29 vs. 17, IQR 12–24 points; *p* < 0.001), SAPS II (58, IQR 45–71 vs. 46, IQR 30–58 points, *p* = 0.001), and SOFA (12, IQR 9–14 vs. 8, IQR 6–11 points; *p* < 0.001) scores at ICU admission. Selected demographic and clinical data are presented in Table 1.

There was no significant correlation between P-SEP and creatinine (*p* = 0.1). P-SEP was not statistically different between patients with AKI and those with normal renal function (11.9, IQR 6.7–16.7 vs. 8.8, IQR 4.3–13.4 ng/mL; *p* = 0.1) (Figure 1).

Significantly higher values of PCT, IL-6, and CRP (*p* < 0.05 for all) were observed in patients who were diagnosed with acute kidney injury (AKI) on admission (Figure 2) and in patients who later started continuous renal replacement therapy (CRRT).

There was no difference in P-SEP, CRP, PCT, and IL-6 values between patients who survived and those who died (*p* > 0.05 for all). P-SEP, CRP, PCT, and IL-6, determined at ICU admission, did not accurately predict the risk of death in ROC curve analysis (*p* > 0.05 for all) (Figure 3).

Definitive confirmation of the location of the focus of bacterial infection by microbiological testing was obtained in only 43 (49%) patients. ICU mortality was higher in patients with a microbiologically confirmed source of infection (45% vs. 25%). The most common foci of infection were the abdominal cavity (*n* = 21, 24%), respiratory tract (*n* = 11, 13%), and urinary tract (*n* = 10, 11%). Soft tissue infection was confirmed in 1 patient (1%). P-SEP, CRP, PCT, and IL-6 were significantly higher in patients with positive microbiological findings, respectively. P-SEP 5.7 (2.7–11.5) vs. 12.3 (8.4–14.8) ng/mL, CRP 131 (78–209) vs. 238 (121–320) mg/L, PCT 1.7 (0.8–4.4) vs. 12.3 (2.9–37.9) ng/mL, IL-6 67 (31–162) vs. 302 (71–752) pg/mL (*p* < 0.001 for all) (Figure 4A–D). P-SEP predicted a positive microbiological result with a cutoff point >5.7 ng/mL (AUC = 0.72, 95%CI 0.62–0.81; *p* < 0.001) with sensitivity of 93% and specificity of 51% (Figure 5).

### Patients with a Positive Microbiological Test

In a retrospective analysis in a subgroup of patients with a positive microbiological test (*n* = 43, 49%), CRP and PCT accurately predicted the risk of death. However, the area under the curve (AUC) in the ROC curve analysis was not clinically satisfactory. In contrast, IL-6 and P-SEP had no significant predictive accuracy in this subgroup (*p* < 0.05 for both) (Table 2).

## 4. Discussion

In our study, we showed that at the time of ICU admission, none of the inflammatory parameters were a good predictor of death in patients with suspected sepsis. Patients with bacteria identified in microbiological tests taken on ICU admission had higher baseline inflammatory parameters, including P-SEP. Mortality rates were higher in patients with an identified pathogen compared to patients with negative microbiological findings.

The lack of predictive accuracy of P-SEP in predicting death in patients with sepsis can be attempted to be explained based on pathophysiological knowledge. Infection leads to the initiation of an innate and acquired immune response. One of the biological defense mechanisms is the release of neutrophil extracellular traps (NETs). P-SEP is produced by phagocytosis of NETs by monocytes. Given the small number of monocytes and their short half-life in the blood, P-SEP can also be released by specific macrophages phagocytizing pathogen fragments [12]. The release of increased amounts of P-SEP into the blood only indicates the phagocytosis of bacterial fragments and the activation of innate effector cells of the immune system [13]. In turn, dysregulation of the immune response is responsible for the increased mortality in sepsis [14]. P-SEP is not a marker of this dysregulation.

Previous studies have demonstrated the usefulness of P-SEP determination in differentiating sepsis from non-septic conditions in burns [15], exacerbation of chronic obstructive pulmonary disease [16], or cholelithiasis [17], among others. Stoma I et al. used P-SEP to rule out bacterial infection and thus increase the likelihood of diagnosing fungal infection in hematological patients [18].

Although P-SEP is presented in studies as a new infectious biomarker, in our study, it did not show superiority over PCT and IL-6 previously used in clinical practice. The results obtained are consistent with those of other authors. A meta-analysis by Kondo Y et al. that included 3012 patients from 19 studies provided evidence that the diagnostic accuracy of PCT and P-SEP in detecting bacterial infection was similar (AUROC 0.84 [95% CI 0.81 to 0.87] and 0.87 [95% CI 0.84 to 0.90], respectively). However, the included studies had significant heterogeneity [19]. A multi-marker approach, including combining several infectious parameters with the SOFA score in the Kim et al. study, better predicted 30-day mortality than the SOFA score alone and each infectious parameter separately [20].

We showed no statistical difference in P-SEP values between patients with normal renal function and those with AKI. However, median P-SEP values in AKI were higher in our population. The association of elevated P-SEP values with AKI is known from previous studies. A positive correlation with P-SEP has previously been observed not only for creatinine and GFR but also for cystatin-C, which suggests that renal function should be considered in the assessment of P-SEP, independent of our single-center observation [21], for example, in the analysis of confounding factors. In the study by Kim SY et al., P-SEP accurately predicted the risk of AKI in patients with sepsis in the emergency department. P-SEP values were significantly higher in stage 3 AKI (as defined by KDIGO) compared to stages 1 and 2 (*p* = 0.001) [22]. Classical definitions of AKI require observation of trends in clinical and laboratory parameters over time, which can determine treatment outcomes in critical illness [23]. P-SEP, as an early marker of increased risk of AKI, would allow the earlier identification of patients who may need CRRT, which is not available at every center. To date, few studies have assessed the predictive accuracy of death in sepsis in patients with renal impairment. In the study by Lee GB et al., the predictive accuracy of death in septic kidney injury requiring CRRT, assessed as AUC, was 0.79 (95%CI 0.65–0.95), which was better than the APACHE II score (AUC 0.63; 95%CI 0.45–0.83) and the SOFA score (AUC 0.69; 95%CI 0.52–0.88) [24]. In a study by Kim SY et al. involving patients with sepsis with COVID-19 etiology (coronavirus disease 2019), both P-SEP and PCT predicted AKI and 30-day in-hospital mortality. The P-SEP cutoff value for predicting mortality was 865 pg/mL, with a sensitivity of 85% and specificity of 76%. P-SEP levels were significantly higher in patients with AKI (801, IQR 442–1589 vs. 447, IQR 239–658 pg/mL; *p* < 0.001) [25].

### Study Limitation

Our study has several limitations. First, it is a single-center study, and extrapolating the results to other populations may not be appropriate. Second, we did not perform serial measurements of P-SEP during hospitalization. However, it should be emphasized that the purpose of the study was to answer a specific clinical question—whether a physician can use P-SEP at the time of ICU admission as a marker of bacterial etiology of organ failure. This reflects real-world clinical practice, in which a physician often cannot defer the decision to initiate antibiotic therapy until a series of measurements have been taken because any delay in antibiotic therapy for sepsis increases mortality [1]. Third, comparing multiple markers in a single study poses risks related to the different kinetics of changes in marker concentrations from the onset of infection. However, a small study involving five healthy volunteers showed that after stimulation with bacterial LPS, an increase in P-SEP occurred concurrently with an increase in IL-6 and was seen as early as one hour after LPS exposure [9]. We did not identify similar studies comparing P-SEP with other markers. Fourth, the extent of microbiological testing was left to the individual physician’s discretion in each case. The possibility exists that we did not identify all of the potential sources of infection by falsely considering sepsis as NIOF. Fifth, a comparison of the results obtained with those of other authors may not be reliable due to the changing definition of SIRS and sepsis and its various forms over the years. Sixthly, several reports have reported significant differences in P-SEP values depending on the age of patients. Previously, Chenevier-Gobeaux et al. [9] reported P-SEP differences in healthy patients. At the same time, a new publication by Shimoyama Y et al. assessed a group of 83 adult sepsis patients and divided them into two groups according to age (under and over 75 years). P-SEP was shown to be a more useful predictor of septic AKI and adult acute respiratory distress syndrome in a population >75 years of age than in younger patients with sepsis [26]. Our study was not designed to compare P-SEP according to age, and the group of patients older than 75 years was small (*n* = 14, 16%) to draw conclusions from. Further prospective studies on a large population of seniors should provide an answer to the question of what P-SEP values may be a predictor of endpoints in elderly patients with sepsis.

## 5. Conclusions

In patients with suspected sepsis admitted to the Intensive Care Unit, presepsin does not accurately predict the risk of in-hospital death, but it can predict a positive microbiological culture.

## Figures and Tables

**Figure 1 biomedicines-12-02313-f001:**
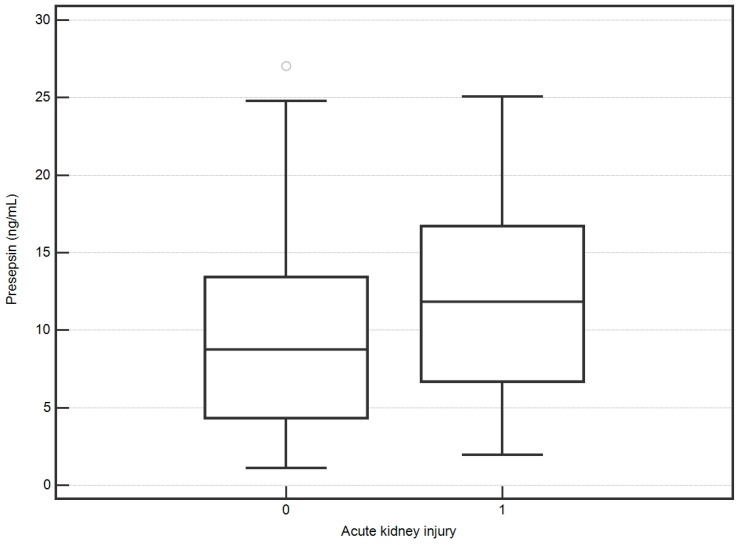
Presepsin values and acute kidney injury.

**Figure 2 biomedicines-12-02313-f002:**
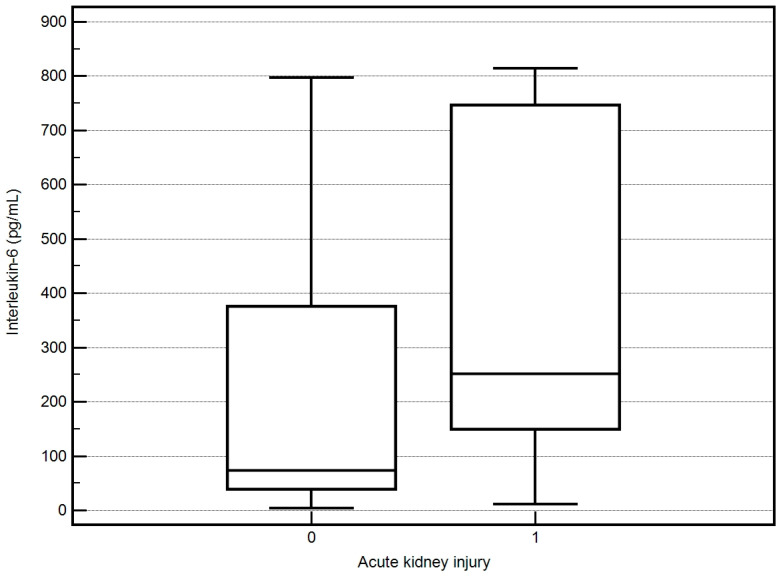
IL-6, PCT, and CRP values and acute kidney injury.

**Figure 3 biomedicines-12-02313-f003:**
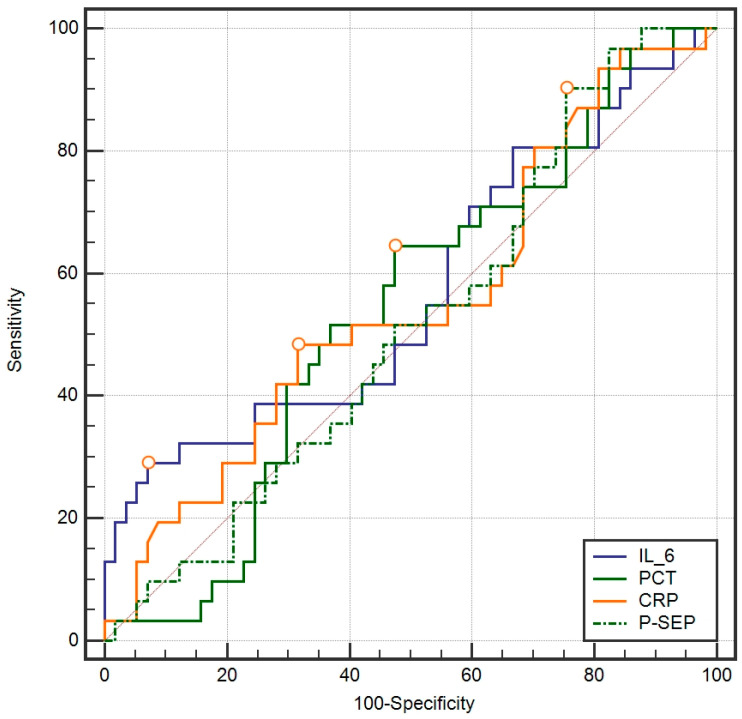
Predictive accuracy of death for IL-6, PCT, CRP, and P-SEP.

**Figure 4 biomedicines-12-02313-f004:**
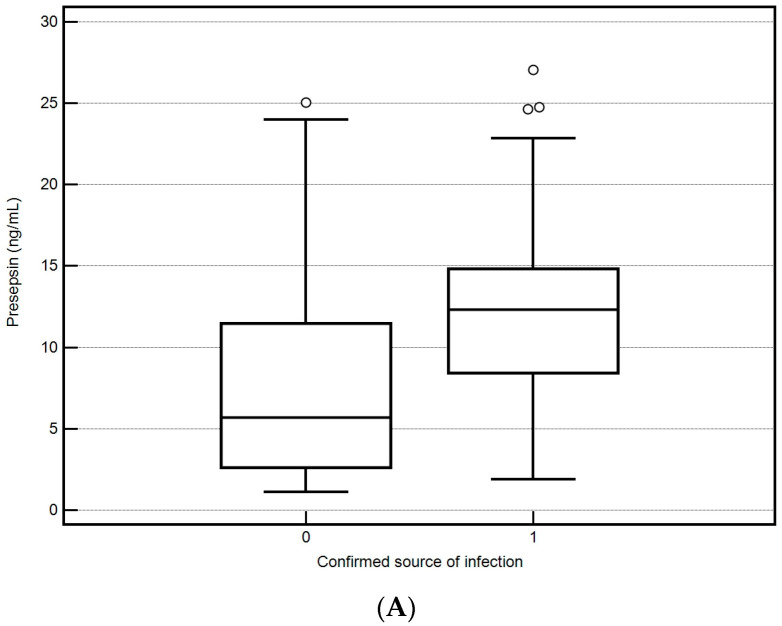
(**A**–**D**) Values of inflammatory parameters and the result of microbiological examination.

**Figure 5 biomedicines-12-02313-f005:**
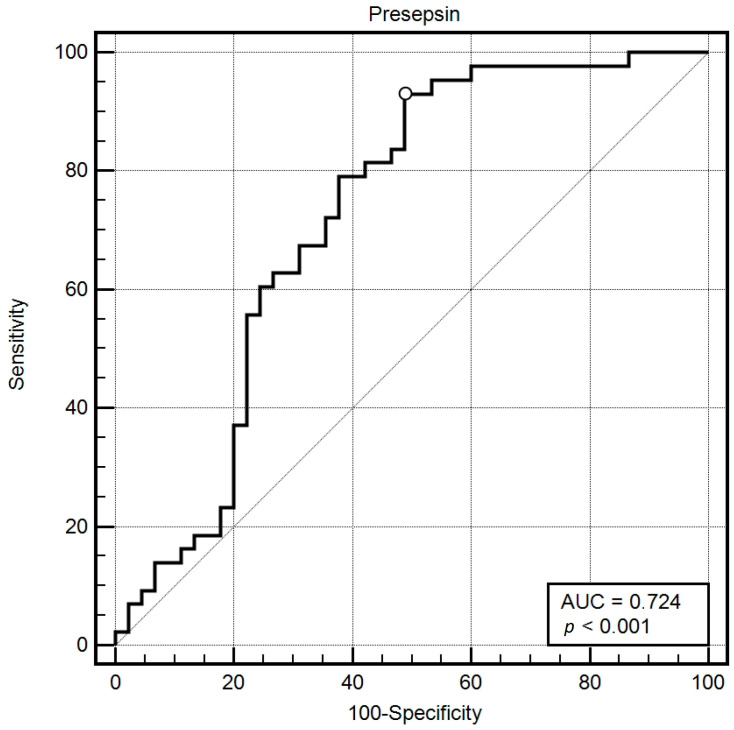
Predictive accuracy of presepsin in predicting a positive microbiological test result.

**Table 1 biomedicines-12-02313-t001:** Selected demographic and clinical data.

Value	All	ICU Discharge (1)	Death before ICU Discharge (2)	*p*-Value (1 vs. 2)
Male sex, *n* (%)	48 (56%)	30 (35%)	18 (21%)	0.6
Age, Me (IQR)	65 (51–72)	64 (45–70)	70 (55–74)	0.1
Predictive scales				
APACHE II, pts, Me (IQR)	20 (15–26)	17 (12–25)	25 (18–29)	0.002
SAPS II, pts, Me (IQR)	49 (36–66)	45 (30–57)	57 (44–71)	0.001
SOFA, pts, Me (IQR)	9 (7–12)	8 (6–11)	12 (9–14)	0.001
Acute Kidney Injury, *n* (%)	22 (26%)	12 (14%)	10 (12%)	0.3
Renal replacement therapy, *n* (%)	11 (13%)	6 (7%)	5 (6%)	0.5
Positive microbiological test, *n* (%)	43 (49%)	24 (28%)	19 (22%)	0.1
Location of infection *, *n* (%)				0.6
Abdominal	21 (24%)	10 (12%)	11 (13%)	
Respiratory system tract	11 (13%)	5 (6%)	6 (7%)	
Urinary system tract	10 (11%)	5 (6%)	5 (6%)	
Soft tissues	1 (<1%)	-	1 (<1%)	

Me (IQR), median (interquartile range); APACHE II, Acute Physiology And Chronic Health Evaluation II; SAPS II, Simplified Acute Physiology Score; SOFA, Sequential Organ Failure Assessment; * based on microbiological test.

**Table 2 biomedicines-12-02313-t002:** Predictive accuracy of death for selected parameters in a group of patients with positive microbiological tests.

Parameter	AUC (95% CI)	Sensitivity (%)	Specificity (%)	Cut-Off	*p*-Value
Interleukin-6	0.60 (0.45–0.71)	52	83	>670 (pg/mL)	0.3
Presepsin	0.53 (0.37–0.68)	89	29	>16.6 (ng/mL)	0.7
C-reactive protein	0.67 (0.51–0.80)	89	42	>311 (mg/L)	0.04
Procalcitonin	0.69 (0.53–0.82)	53	79	>3.64 (ng/mL)	0.02

AUC, area under curve; 95%CI (95% confidence interval).

## Data Availability

For legal reasons data are available from the authors of the study and can be made available upon request to other scientific institutions.

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
