# Peer review of "Presepsin Does Not Predict Risk of Death in Sepsis Patients Admitted to the Intensive Care Unit: A Prospective Single-Center Study"

_biomedicines, 2024, doi:10.3390/biomedicines12102313_

Round 1

Reviewer 1 Report

Comments and Suggestions for Authors

The study presents an interesting exploration into the use of presepsin as a biomarker for predicting mortality risk in sepsis patients admitted to the ICU. While the topic is relevant and contributes to the ongoing research in sepsis biomarkers, I find that the novelty of the study is limited, as similar research has been conducted previously. The small sample size further restricts the generalizability of the findings.

Moreover, I have reservations regarding the main finding that presepsin can predict a positive microbiological culture. The ROC curve analysis provided in the paper does not support presepsin's predictive value convincingly. The AUC values indicate a lack of strong predictive power, which suggests that presepsin may not be as reliable in this context as the study suggests.

Additionally, if the paper is considered for publication, I would recommend revising the title to better reflect the negative findings. Emphasizing the limited predictive value of presepsin would provide a more accurate representation of the study's outcomes.

Thank you for the opportunity to review this paper. I believe that with a larger sample size and additional analysis, the study could potentially offer more robust conclusions.

Comments on the Quality of English Language

The english is fine. 

Author Response

Thank you for evaluating our paper.

As suggested by the reviewer, the title of the article has been changed, which clearly indicates the lack of predictive accuracy of presepsin in predicting death.

Regarding the reviewer's comment regarding the ROC analysis for predicting microbiological positivity based on presepsin - the ROC analysis for predicting microbiological positivity was not presented anywhere in the article presented to the reviewer. Figure 3 shows the lack of predictive accuracy of death for all biomarkers included in the study. In contrast, Table 2 shows the predictive accuracy of death in the group of patients with a positive microbiological result. We decided to perform an additional analysis as suggested by the reviewer on the predictive accuracy of presepsin in predicting a positive microbiological test result. The AUC ROC was 0.72 (95%CI 0.62 - 0.81), p<0.001. The cut-off point for predicting a positive microbiological test result was P-SEP >5.7 ng/mL. The results obtained are appended to the manuscript.

Thank you again for evaluating our work. We hope that it will receive reviewer approval once corrections have been incorporated.

Best regards,

Authors

Reviewer 2 Report

Comments and Suggestions for Authors

The manuscript titled "Diagnostic accuracy of presepsin in predicting the risk of death in patients with sepsis admitted to the Intensive Care Unit: a prospective single-center study" provides an understanding exploration into the role of presepsin (P-SEP) as a biomarker for sepsis outcomes in critically ill patients. The study aimed to evaluate the efficacy of P-SEP in predicting mortality in ICU-admitted sepsis patients while comparing it with other biomarkers such as C-reactive protein (CRP), procalcitonin (PCT), and interleukin-6 (IL-6).

The introduction clearly establishes the significance of sepsis as a leading cause of mortality in ICU.

The study is well designed being prospective and included adult ICU patients who met the SEPSIS-3 criteria. The primary endpoint was ICU mortality, with secondary endpoints pointing on microbiological confirmation of infection.

The results are well structured indicating no significant difference in the levels of P-SEP, CRP, PCT, and IL-6 between survivors and non-survivors. P-SEP did not predict ICU mortality in the ROC curve analysis but the inflammatory markers, including P-SEP, were significantly higher in patients with confirmed infections.

In the discussion section, the authors admitted the limitations of presepsin in predicting mortality but highlighted its role in identification of bacterial infections. The authors emphasized the need for further large studies to enhance the clinical utility of presepsin.

In conclusion, the study found that presepsin is not an accurate predictor of mortality in septic patients but may still be useful in predicting positive microbiological cultures. The manuscript disclosed in a correct approach the limitations of the study and future researches.

The tables and figures summarize data of the study providing a clear and accessible presentation of the study's findings.

The references are relevant, reflecting a comprehensive review of the literature.

The manuscript is a well-structured with a clear and logical flow. Overall, the manuscript is a strong contribution to the literature.

Therefore, I recommend that the paper can be published in the current form.

Author Response

Thank you very much for taking the time to review our work.

Thank you for your positive review of our work.

With best regards,

Authors

Reviewer 3 Report

Comments and Suggestions for Authors

This prospective single-center study provides some information about the predictive value of presepsin in sepsis patients. The authors concluded that while presepsin cannot predict the risk of in-hospital death, it can predict the presence of a microbiological infection. While the research holds some scientific value, there are several issues that should be addressed:

Presepsin is primarily released by monocytes after phagocytosis of neutrophil extracellular traps (NETs). It is strongly associated with the presence of infection but not with the severity of infection. As a result, prepepsin cannot serve as a reliable marker for predicting mortality in sepsis. This study confirms this hypothesis, but it also limits the paper's impact as the finding is not unexpected. This should be discussed.

     The figures appear to have been generated without a consistent format. For example, Figure 1 includes a round, transparent dot, while Figure 2 uses a completely different style, which creates confusion. Figures 4A-D also suffer from a similar inconsistency.

   In Figure 4, the mean and 95% CI values should not be displayed in the figure itself. Instead, these statistical values should be included in the Results section.

Author Response

We would like to thank you for your thorough review of our work and your valuable comments.

As suggested by the reviewer, we have standardised the formatting of the figures. 

We have also expanded the discussion to include an issue suggested by the reviewer.

We hope that the manuscript presented in its present form will meet the reviewer's appreciation.

Best regards,

Authors

Round 2

Reviewer 1 Report

Comments and Suggestions for Authors

Thank you very much for reviewing the manuscript, and I apologize for the mix-up. I had assumed that Table 2 represented the prediction of a positive blood culture. However, the manuscript should be revised in terms of structure, and there are still some spelling errors. Why could no additional cases be included?

Comments on the Quality of English Language

The english should be revised. There are several spelling errors. In addition, the manuscript should be better structured.

Author Response

Thank you for your comments.

(1) The manuscript has been corrected by a native speaker for linguistic errors

(2) In response to the question about the number of participants in the study, we would like to inform you that we were unable to include additional patients because the planned recruitment period for participants had ended. The study was prospective in nature. The number of presepsin assays planned was limited by the availability of ELISA reagents.

Thank you again for taking the time to review our work.